# MEMO: MEMORY-GUIDED AND EMOTION-AWARE TALKING VIDEO GENERATION

## ABSTRACT

Advances in video diffusion models have unlocked the potential for realistic audio-driven talking video generation. However, it is still highly challenging to ensure seamless audio-lip synchronization, maintain long-term identity consistency, and achieve natural expressions aligned with the audio in generated talking videos. To address these challenges, we propose **M**emory-guided **EMO**tion-aware diffusion (MEMO), an end-to-end audio-driven portrait animation approach to generate identity-consistent and expressive talking videos. Our approach is built around two key modules: (1) a memory-guided temporal module, which enhances long-term identity consistency and smooth motion by developing memory states that store information from all previously generated frames and guide temporal modeling through linear attention; and (2) an emotion-aware audio module, which replaces traditional cross attention with multi-modal attention to enhance audio-video interaction, while detecting emotions from the audio to refine facial expressions via emotion adaptive layer norm. Moreover, MEMO is trained on a large-scale, high-quality dataset of talking head videos without relying on facial inductive biases such as face landmarks or bounding boxes. Extensive experiments demonstrate that MEMO generates more realistic talking videos across a wide range of audio types, surpassing state-of-the-art talking video diffusion methods in human evaluations in terms of emotion-audio alignment, identity consistency and overall quality, respectively.

## 1 INTRODUCTION

Audio-driven talking video generation (Prajwal et al., 2020; Tian et al., 2024; Xu et al., 2024b) has gained significant attention due to its broad impact on areas like virtual avatars, digital content creation, and real-time communication, offering transformative possibilities in entertainment, education, and e-commerce. However, compared to text-to-video generation (Guo et al., 2023; Rombach et al., 2022; Ramesh et al., 2022) or image-to-video generation (Blattmann et al., 2023), audio-driven talking video generation presents unique challenges. It requires not only generating synchronized lip movements and realistic head motions from audio, but also preserving the long-term identity consistency of the reference image and producing natural expressions that align with the emotional tone of the audio. Successfully balancing these demands while ensuring generalization across diverse audio inputs and reference images makes this task highly challenging.

Recent advances in video diffusion models (Tian et al., 2024; Xu et al., 2024a; Chen et al., 2024) have enabled more realistic audio-driven talking video generation. Most existing methods use cross-attention mechanisms to incorporate audio to guide video generation and typically condition on past generated 2-4 frames to improve identity consistency and motion smoothness (Tian et al., 2024; Xu et al., 2024a). Sometimes, they incorporate a static emotion label to specify the emotion of the generated video (Xu et al., 2024b; Tan et al., 2024). However, these approaches face challenges with audio-lip synchronization, maintaining long-term identity consistency, and achieving natural expressions aligned with the audio. Cross-attention mechanisms rely on fixed audio features, limiting audio-video interaction and coherence, while conditioning on a limited number of past frames can lead to error accumulation, especially when those frames contain artifacts (cf. Figure 1). Additionally, using static emotion labels can result in facial expressions that fail to capture the dynamic emotional shifts inherent in audio. Consequently, these methods may struggle with lip-audio synchronization, expression-audio alignment, and long-term identity preservation.

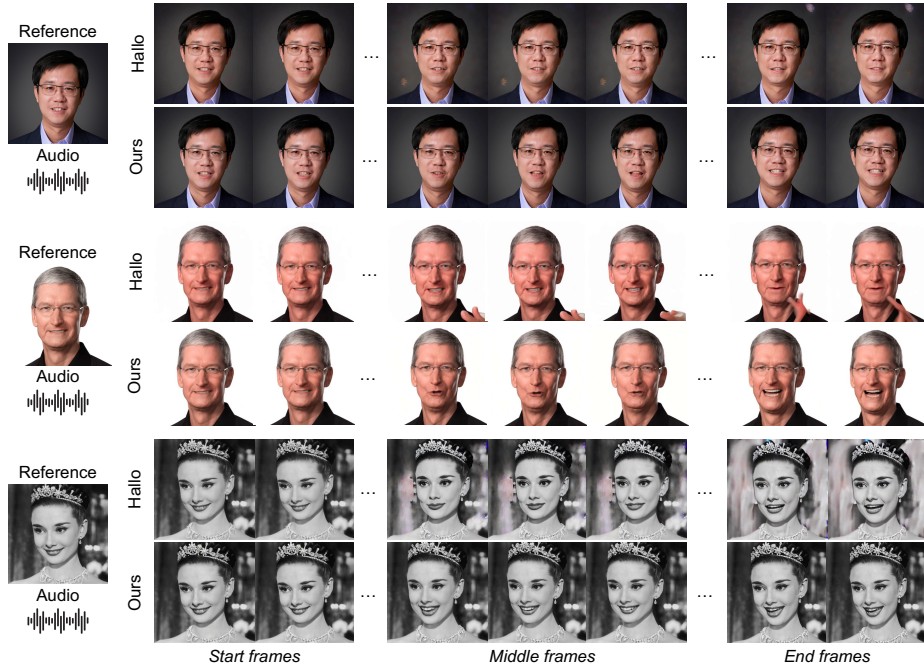

Figure 1: Our MEMO generates talking videos with improved identity consistency, audio-lip alignment, and motion smoothness. In contrast, existing diffusion-based methods (*e.g.*, Hallo (Xu et al., 2024a)) are prone to error accumulation during auto-regressive generation, especially when the generated last 2-4 frames used as temporal conditions contain artifacts, leading to inconsistent identity.

In this paper, we propose **M**emory-guided **EMO**tion-aware diffusion (MEMO), an end-to-end audio-driven portrait animation approach. As shown in Figure 2, MEMO is built around two key modules: (1) a memory-guided temporal module and (2) an emotion-aware audio module. To ensure consistent facial identity and smooth transitions across long-term videos, MEMO develops a memory-guided temporal module that maintains memory states across all previously generated frames. This allows the model to use long-term motion information to guide temporal modeling through linear attention, resulting in more coherent facial movements and mitigating the error accumulation issue that may occur in existing diffusion methods (cf. Figure 1). Moreover, to improve audio-lip synchronization and align facial expressions with the emotional tone of the audio, MEMO introduces an emotion-aware audio module. This module replaces the traditional cross-attention audio module in previous diffusion methods with a more dynamic multi-modal attention mechanism, enabling better interaction between audio and video during the diffusion process. By detecting subtle emotional cues from the audio, this module further refines facial expressions through emotion adaptive layer norm, enabling the generation of expressive and emotionally aligned talking videos.

Extensive quantitative results and human evaluations demonstrate that our approach consistently outperforms state-of-the-art methods in overall quality, audio-lip synchronization, expression-audio alignment, identity consistency, and motion smoothness (cf. Table 1 and Figure 6). Additionally, diverse qualitative results highlight MEMO's strong generalization across various types of audio, including speech, singing, rap, and multiple languages, further showcasing the effectiveness of our method. Lastly, ablation studies further validate the distinct contributions of the emotion-aware audio module, which significantly improves audio-lip alignment and expression naturalness, and the memory-guided temporal module, which enhances long-term identity consistency and motion smoothness.

In summary, our contributions are threefold: (1) MEMO is the first to leverage motion information from all past frames to guide temporal modeling in diffusion-based talking video generation, effectively improving long-term identity consistency and motion smoothness; (2) unlike previous methods, MEMO dynamically detects the emotion in audio and incorporates it into audio-video interaction, improving lip-audio synchronization and expression-audio alignment in talking videos; (3) we introduce a new data processing pipeline to obtain high-quality talking head data, which is crucial for diffusion model training and will benefit future research in talking video generation.

## 2 RELATED WORK

**Audio-driven talking head generation.** Audio-driven talking head generation aims to synthesize realistic and synchronized talking videos given an audio clip and a reference image. Early approaches only focused on learning audio-lip mapping while keeping other facial attributes static (Suwajanakorn et al., 2017; Chen et al., 2018; Prajwal et al., 2020; Cheng et al., 2022; Yin et al., 2022). These methods could not capture comprehensive facial expressions and natural head movements. To improve realism, later research leveraged intermediate motion representations, *e.g.*, landmark coordinates, 3D facial mesh, and 3D morphable models, and decomposed the generation process into two stages, *i.e.*, audio-to-motion and motion-to-video (Zhou et al., 2020; Sun et al., 2023; Zhang et al., 2023b; Wang et al., 2024; Chen et al., 2024; Wei et al., 2024). The typical issue of these methods is the bottleneck of the intermediate representations, which limits the expressiveness and realism of the generated videos. Recent end-to-end methods can generate vivid portrait videos (Tian et al., 2024; Xu et al., 2024a) by fine-tuning pre-trained text-to-video (T2V) models, but they struggle to generalize to out-of-distribution (OOD) scenarios and need specific modules (*e.g.*, face locator) to constrain head stability, which hinders more natural head motions. Similar issues exist in the methods that learned a specific face latent space (He et al., 2023; Ma et al., 2023; Zhang et al., 2023a; Xu et al., 2024b). Furthermore, most of these methods use 2-4 past frames to generate long videos auto-regressively, which may lead to error accumulation over time and fail to preserve identity when generating long-term videos. In contrast, our work does not depend on any facial inductive biases, which unlocks the possibilities for generating more expressive head motions of talking videos. Moreover, our method improves long-term identity consistency and mitigates error accumulation via memory-guided linear attention. Besides, unlike previous diffusion-based methods that used a cross-attention mechanism to integrate audio features, our method enhances the lip-audio synchronization and expression-audio alignment based on a newly developed emotion-aware multi-modal diffusion. The most related concurrent work to our memory module is Loopy (Jiang et al., 2024), which use a temporal segment module to model cross-clip relationships, but it only considers the representative motion frames in other temporal segments. In contrast, our memory-guided temporal module allows MEMO to utilize all past frames to provide more comprehensive temporal guidance for motion and identity. More related studies of diffusion models are provided in Appendix C.

## 3 PRELIMINARIES

**Problem statement.** Given a reference image and audio as inputs, audio-driven talking video generation (Prajwal et al., 2020; Tian et al., 2024) aims to output a vivid video that closely aligns with the input audio and authentically replicates real human speech and facial movements. This task is particularly challenging because it requires seamless audio-lip synchronization, realistic head movements, long-term identity consistency, and natural expressions that align with the audio. Most existing diffusion-based approaches (Tian et al., 2024; Xu et al., 2024a; Chen et al., 2024) struggle with issues such as error accumulation, inconsistent identity preservation over time, limited audio-lip synchronization, unnatural expressions, and poor generalization.

**Latent diffusion models and rectified flow loss.** Our method is built upon the Latent Diffusion Model (LDM) (Rombach et al., 2022), a framework designed to efficiently learn generative processes in a lower-dimensional latent space rather than directly operating on pixel space. During training, LDM first employs a pre-trained encoder $\mathcal{E}(\cdot)$ to map high-dimensional images into a compressed latent space, producing latent features $z_0 = \mathcal{E}(I)$. Then, following the principles of Denoising Diffusion Probabilistic Models (DDPM) (Ho et al., 2020), Gaussian noise $\epsilon$ is progressively added to the latent features over $t$ discrete timesteps, resulting in noisy latent features $z_t = \sqrt{\alpha_t} z_0 + \sqrt{1 - \alpha_t} \epsilon$, where $\alpha_t$ is a variance schedule controlling how much noise is added. The diffusion model is then trained to reverse this noise-adding process by taking the noisy latent representation $z_t$ as input and predicting the added noise $\epsilon$. The objective function for training can be expressed as: $\mathcal{L} = \mathbb{E}_{z_t, c, \epsilon \sim \mathcal{N}(0,1), t}[\|\epsilon - \epsilon_\theta(z_t, t, c)\|_2^2]$, where $\epsilon_\theta$ represents the noise prediction made by the U-Net network, and $c$ represents conditioning information such as audio, or motion frames in the context of talking video generation. Recently, Stable Diffusion 3 (SD3) (Esser et al., 2024) introduced a refinement to this process by incorporating rectified flow loss, which modifies

Figure 2: Overview of MEMO.

the traditional DDPM objective to:

$$\mathcal{L} = \mathbb{E}_{z_t, c, \epsilon \sim \mathcal{N}(0,1), t}[\lambda(t)\|\epsilon - \epsilon_\theta(z_t, t, c)\|_2^2], \tag{1}$$

where $\lambda(t) = 1/(1-t)^2$ and $z_t$ is reparameterized using linear combination as $z_t = (1-t)z_0 + t\epsilon$. This formulation leads to both better training stability and more efficient inference. In light of these advantages, we adopt the rectified flow loss from SD3 in our training.

## 4 METHOD

As illustrated in Figure 2, MEMO is an end-to-end audio-driven diffusion model for generating identity-consistent and expressive talking videos. Similar to previous diffusion-based approaches (Tian et al., 2024; Xu et al., 2024a), MEMO is built around two main components: a Reference Net and a Diffusion Net. The main contributions of MEMO lie in two key modules within the Diffusion Net: the **memory-guided temporal module** (cf. Section 4.1), and the **emotion-aware audio module** (cf. Section 4.2), which work together to achieve superior audio-video synchronization, long-term identity consistency, and natural expression generation. In addition, MEMO introduces a new data processing pipeline (cf. Section 4.4) for acquiring high-quality talking head videos, along with a decomposed training strategy (cf. Section 4.3) to optimize diffusion model training.

### 4.1 MEMORY-GUIDED TEMPORAL MODULE

Most existing diffusion-based approaches (Tian et al., 2024; Xu et al., 2024a; Chen et al., 2024) typically generate talking videos in an auto-regressive manner, dividing the audio into clips corresponding to 12-16 frames and using the past 2-4 generated frames to condition the generation of the next video clip. They concatenate the past frame features with the current noisy latent features along the temporal dimension and apply temporal self-attention to model the sequential information. While this approach can model short-term dependencies, it often struggles with maintaining consistency over longer sequences. If artifacts are generated in the past 2-4 conditioned frames, the errors tend to accumulate as the generation progresses, resulting in visual distortions that degrade both identity consistency and audio quality (cf. Figure 1).

Motivated by the idea that leveraging a more complete memory of motion information, rather than relying solely on the most recent 2-4 frames, can provide richer guidance for enhancing identity consistency and motion smoothness, we propose a memory-guided temporal module. The key of this module is memory-guided linear attention, which is designed to improve temporal coherence and maintain consistent facial identity.

**Linear Attention.** Previous approaches use self-attention (Tian et al., 2024; Jiang et al., 2024) to capture temporal information across frames. However, self-attention requires storing all key-value pairs, leading to increasing GPU-memory overhead as the number of memory frames grows, making it impractical to use all motion information. To address this limitation, we replace self-attention with linear attention (Katharopoulos et al., 2020) and include a memory update mechanism into linear

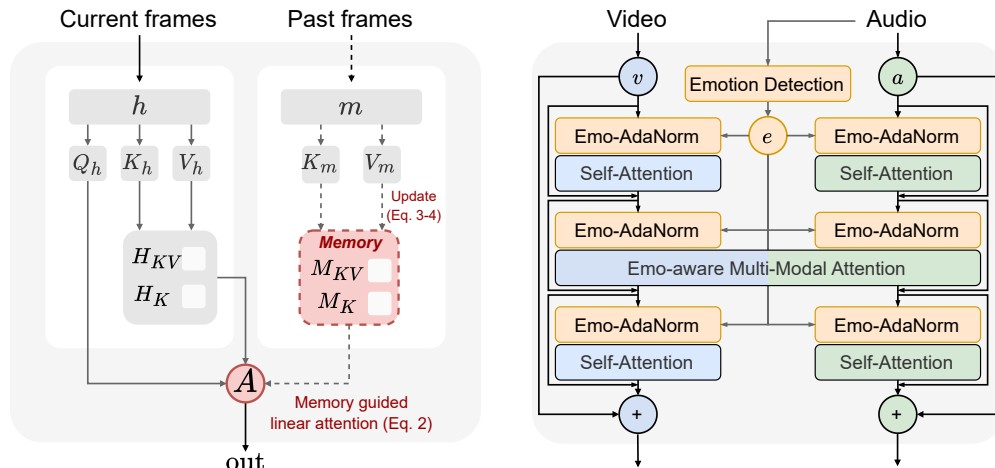

Figure 3: Memory-guided temporal module.      Figure 4: Emotion-aware audio module.

attention to model long-term temporal information efficiently. Denote query as $Q$, key as $K$ and value as $V$. In linear attention, the output for $i$-th frame is

$$\text{out}_i = \frac{\phi(Q_i)^\top \left( \sum_{j=1}^{f} \phi(K_j) V_j^\top \right)}{\phi(Q_i)^\top \sum_{j=1}^{f} \phi(K_j)},$$

where $f$ is the frame number and $\phi$ is an activation function (we use softmax in this work).

**Memory update mechanism.** To incorporate motion information from all past frames to guide video generation, we develop a memory update mechanism. Specifically, let the latent features of past frames as $m \in \mathbb{R}^{f \times d}$ and the latent features of current frames as $h \in \mathbb{R}^{f \times d}$, where $d$ is the dimension of latent features. As shown in Figure 3, linear attention processes these latent features via learnable matrices, which transform them into queries ($Q_h$), keys ($K_h, K_m$), and values ($V_h, V_m$).

To memorize all motion information, we define the memory $M^f$ for the past $f$ frames as two matrices: $M_{KV}^f = \sum_{i=1}^{f} \gamma^i \phi(K_{m,i}) V_{m,i}^\top$ and $M_K^f = \sum_{i=1}^{f} \gamma^i \phi(K_{m,i})$, which occupy constant GPU-memory irrespective of $f$. Here, $\gamma$ is a decay factor ($0 < \gamma < 1$) that modulates the influence of past frames, with more recent frames exerting greater impact, reflected through the exponentiation by $i$. After each generation of $f$ frames, we update the memory $M^f$ by incorporating information from these newly generated frames. In formal, the memory update when adding the latest $a$ frames to the memory with $b$ past frames is as follows:

$$M_{KV}^{a+b} \leftarrow \gamma^a M_{KV}^b + \sum_{j=1}^{a} \gamma^j \phi(K_{h,j}) V_{h,j}^\top, \tag{2}$$

$$M_K^{a+b} \leftarrow \gamma^a M_K^b + \sum_{j=1}^{a} \gamma^j \phi(K_{h,j}). \tag{3}$$

Here, the decay scheme plays a crucial role, since using a unified positional encoding across different clips is infeasible. Instead, we use causal memory decay to provide implicit positional encoding, which enables more effective memory updates for capturing long-term dependencies.

**Memory-guided linear attention.** When generating the current clips, we use the memory to guide the temporal modeling. Let $H_{KV} = \phi(K_h) V_h^\top$ and $H_K = \phi(K_h)$. The output of the memory-guided temporal module is calculated as follows:

$$\text{out} = \frac{\phi(Q_h)^\top (H_{KV} + M_{KV})}{\phi(Q_h)^\top (H_K + M_K)}. \tag{4}$$

## 4.2 EMOTION-AWARE AUDIO MODULE

Existing diffusion-based approaches (Tian et al., 2024; Xu et al., 2024a; Chen et al., 2024) rely on cross-attention mechanisms to integrate audio guidance for video generation, while some methods (Xu et al., 2024b; Tan et al., 2024) further use static emotion labels to generate more emotionally expressive talking videos. However, cross attention relies on fixed audio features, limiting the depth of audio-video interaction during the diffusion process; while static emotion labels cannot capture

the emotional nuances in the audio, leading to facial expressions that do not align naturally with the audio emotions. To address these issues, we develop a new emotion-aware audio module to improve audio-lip consistency and align facial expressions with the audio emotion. As shown in Figure 4, there are two key components: multi-modal attention and Emotion AdaNorm.

**Multi-modal attention**. Our emotion-aware audio module replaces the traditional cross attention with a more dynamic multi-modal attention mechanism. Specifically, cross attention aligns video and audio information by conditioning video features $v$ on audio features $a$. This approach can be formalized as minimizing the loss function $\mathcal{L}_{\theta_{v|a}} = \mathbb{E}_{t,\epsilon \sim \mathcal{N}(0,I)}[\lambda(t)\|\epsilon_\theta(v_t|a) - \epsilon\|_2^2]$. In contrast, we explore multi-modal attention, which jointly processes both video and audio inputs by minimizing the loss function $\mathcal{L}_{\theta_{va}} = \mathbb{E}_{t,\epsilon \sim \mathcal{N}(0,I)}[\lambda(t)\|\epsilon_\theta(v_t, a) - \epsilon\|_2^2]$, enabling better video-audio interaction during the diffusion process.

**Emotion AdaNorm**. We then dynamically detect audio emotions to guide audio-video interaction, using a newly trained emotion detection model. Specifically, the model is trained on a diverse dataset to extract emotion $e$ from audio (see Appendix A for more details), recognizing eight distinct emotions: `angry`, `disgusted`, `fearful`, `happy`, `neutral`, `sad`, `surprised`, and `others`. The detected emotion for each audio clip is then projected into emotion embeddings, which are incorporated into each layer via adaptive layer norm to guide multi-modal attention. This process results in the following emotion-conditioned loss:

$$\mathcal{L}_{\theta_{va|e}} = \mathbb{E}_{t,\epsilon \sim \mathcal{N}(0,I)}[\lambda(t)\|\epsilon_\theta(v_t, a|e) - \epsilon\|_2^2]. \tag{5}$$

During inference, we use classifier-free guidance (Ho & Salimans, 2022) to further enhance the impact of the dynamically detected emotion on the generated output. The emotion-aware output is

$$\tilde{\epsilon}_\theta(v_t, a|e) = (1 + w)\epsilon_\theta(v_t, a|e) - w\epsilon_\theta(v_t, a), \tag{6}$$

where $w$ is the classifier-free guidance scale controlling the influence of the emotion condition. This technique amplifies the emotional cues during inference, allowing MEMO to generate talking videos that are not only synchronized with the speech but also rich in emotional expressiveness.

### 4.3 TRAINING STRATEGY DECOMPOSITION

The model's generative capabilities can be progressively enhanced by dividing the training process into three distinct stages, each with specific objectives.

**Stage 1: Face domain adaptation.** Following (Tian et al., 2024; Xu et al., 2024a; Chen et al., 2024), we initialize the Reference Net and the spatial module of the Diffusion Net with the weights of SD 1.5 (Rombach et al., 2022). In this stage, we adapt the Reference Net, the spatial attention modules of the Diffusion Net, and the original text cross-attention module to the face domain, ensuring these components capture facial features effectively.

**Stage 2: Robust scale-up training.** We then integrate the emotion-aware audio module and memory-guided temporal module into the Diffusion Net. Initially, we perform a warm-up training phase, keeping the modules trained in Stage 1 fixed. After the warm-up, all modules are jointly trained using a fixed number of past frames as memory. Here, since our method generates 16 frames at a time, we set the number of past frames to 16 as temporal context. In this stage, we scale up the dataset to include all collected and processed data for more comprehensive training. However, even after applying our data processing pipeline (cf. Section 4.4) and manual filtering, we found that some noisy data remained, making the diffusion training unstable and leading to biased model optimization. To mitigate this issue, we develop a robust training strategy that filters out data points with loss values suddenly exceeding a specific threshold (0.1 in our case), as the rectified flow loss (cf. Eq. 1) in our method typically converges and fluctuates around 0.03.

**Stage 3: Dynamic past frame training.** In Stage 2, we use a fixed number of 16 past frames to compute memory states. However, during inference, the audio typically spans much longer than 16 frames, meaning the memory must dynamically adapt to longer past frames to avoid a gap between training and inference. To address this, we further introduce dynamic past frame training. During each diffusion training iteration, we randomly select 16, 32, or 48 as the number of past frames, allowing the model to better handle longer memory updates. One might ask why we do not use values larger than 48. This is because, with our memory decay scheme, 48 past frames are sufficient to generalize memory updates over longer sequences, while also keeping computation manageable. In this stage, we train only the audio and temporal modules, while keeping all other modules fixed.

Table 1: Quantitative results of video quality and audio-lip synchronization on the VoxCeleb2 test set and the OOD datasets. MEMO consistently outperforms existing talking video baselines.

| Method | VoxCeleb2 test set | | | OOD dataset | | |
|---|---|---|---|---|---|---|
| | FVD ↓ | FID ↓ | Sync-C ↑ | FVD ↓ | FID ↓ | Sync-C ↑ |
| SadTalker (Zhang et al., 2023b) | 508.8 | 71.4 | 5.7 | 225.3 | 40.9 | 5.6 |
| AniPortrait (Wei et al., 2024) | 291.9 | 37.7 | 3.0 | 266.0 | 37.3 | 3.4 |
| V-Express (Wang et al., 2024) | 445.0 | 46.6 | **7.0** | 316.6 | 45.0 | 5.6 |
| Hallo (Xu et al., 2024a) | 216.9 | 33.2 | 6.9 | 174.4 | 33.0 | 5.9 |
| EchoMimic (Chen et al., 2024) | 396.3 | 81.6 | 4.0 | 202.8 | 43.2 | 5.9 |
| **MEMO (Ours)** | **197.8** | **30.5** | **7.0** | **160.4** | **32.1** | **6.1** |

## 4.4 DATA PROCESSING PIPELINE

We collect a comprehensive set of open-source datasets, such as HDTF (Zhang et al., 2021b), VFHQ (Xie et al., 2022), CelebV-HQ (Zhu et al., 2022), MultiTalk (Sung-Bin et al., 2024), and MEAD (Wang et al., 2020b), along with additional data we collected ourselves. The total duration of these raw videos exceeds 2,200 hours. However, as illustrated in Figure 13 in Appendix B, we find that the overall quality of the data is poor, with numerous issues such as audio-lip misalignment, missing heads, multiple heads, occluded faces by subtitles, extremely small face regions, and low resolution. Directly using these data for model training results in unstable training, poor convergence, and terrible generation quality.

To further obtain high-quality talking head data, we developed a dedicated data processing pipeline for talking head generation. The pipeline consists of five steps: First, we perform scene transition detection and trim video clips to a length of less than 30 seconds. Second, we apply face detection, filtering out videos with no faces, partial faces, or multiple heads, and use the resulting bounding boxes to extract talking heads. Third, we use an Image Quality Assessment model (Su et al., 2020) to filter out low-quality and low-resolution videos. Fourth, we apply SyncNet (Prajwal et al., 2020) to remove videos with audio-lip synchronization issues. Lastly, for partial data, we manually assess the audio-lip synchronization and overall video quality for more accurate filtering. After completing the entire pipeline, the total duration of the processed high-quality videos is approximately 660 hours. We use this processed data for the second and third stages of model training in Section 4.3.

## 5 EXPERIMENTS

### 5.1 EXPERIMENTAL SETUP

**Evaluation benchmarks.** We create two datasets to evaluate MEMO's performance and generalization capabilities. We use the VoxCeleb2 (Nagrani et al., 2020) test set, which contains videos of various celebrities. We select 46 individuals and sample 10 clips per person, resulting in a total of 460 clips. To further evaluate out-of-distribution (OOD) generalization, we create an OOD dataset with 300 video clips across a more diverse set of audios, backgrounds, ages, genders, languages, *etc.*

**Evaluation metrics.** We adopt a suite of metrics to evaluate the overall quality and audio-lip synchronization of the generated videos. The Fréchet Video Distance (FVD) (Unterthiner et al., 2019) measures the distance between the distributions of real and generated videos, providing an assessment of overall video quality. The Fréchet Inception Distance (FID) (Heusel et al., 2017) evaluates the quality of individual frames by comparing feature distributions extracted from a pre-trained model. SyncNet Confidence (Sync-C) (Chung & Zisserman, 2017) measures audio-lip synchronization using a pre-trained discriminator model.

**Baselines.** We compare our method against several state-of-the-art baselines, including two-stage methods with intermediate representations and end-to-end audio-to-video diffusion methods. V-Express (Wang et al., 2024) and EchoMimic (Chen et al., 2024) are two-stage methods using intermediate representations like landmarks, while Hallo (Xu et al., 2024a) is a recent end-to-end diffusion model using hierarchical face masks to integrate audio information. More implementation details of MEMO are put into Appendix D.

378
379
380
381
382
383
384
385
386
387
388
389
390
391
392
393

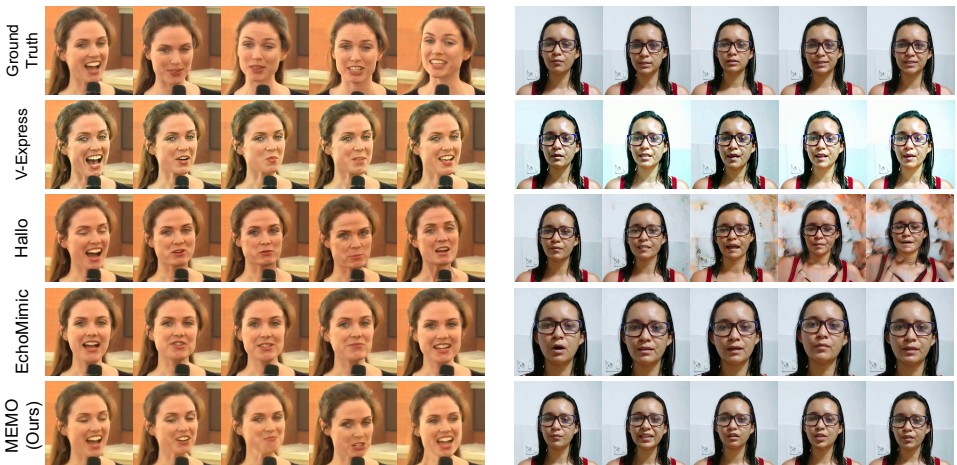

Figure 5: Visualization of generated videos on VoxCeleb2 (left) and the OOD dataset (right).

394
395
396
397
398
399
400
401
402
403
404
405
406
407
408
409
410
411
412
413

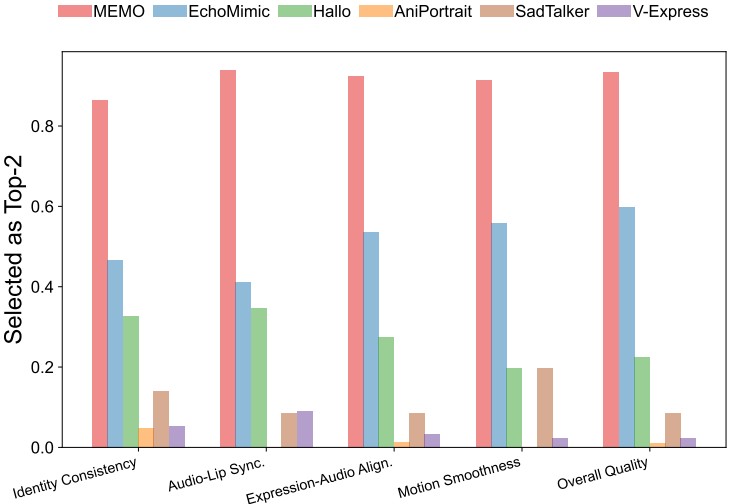

Figure 6: Human preferences among MEMO and baselines.

414
415
416
417
418
419
420
421
422
423
424
425
426
427
428
429
430
431

## 5.2 QUANTITATIVE RESULTS

**Performance on VoxCeleb2 test set and OOD dataset.** Table 1 summarizes the quantitative results on VoxCeleb2 and our collected OOD dataset. In the VoxCeleb2 test set, our method consistently outperforms all baselines across FVD, FID, and Sync-C metrics, indicating better video quality and audio-lip synchronization. Meanwhile, MEMO maintains robust performance in OOD datasets compared to baselines, demonstrating improved generalization to unseen identities and challenging reference images and audios.

**Human evaluation.** To better benchmark the quality of generated talking videos, we conduct human studies based on five subjective metrics in several challenging scenarios, *e.g.*, singing, rap, and multilingual talking video generation. Specifically, our analyses are based on the overall quality, motion smoothness, expression-audio alignment, audio-lip synchronization, and identity consistency. As shown in Figure 6, our method achieves the highest scores across all criteria in human top-2 choice evaluations. Specifically, MEMO is selected as the best case in 93.3%, 91.4%, 92.4%, 93.8%, and 86.6% of the samples for the five metrics, respectively. This further demonstrates the effectiveness of our approach.

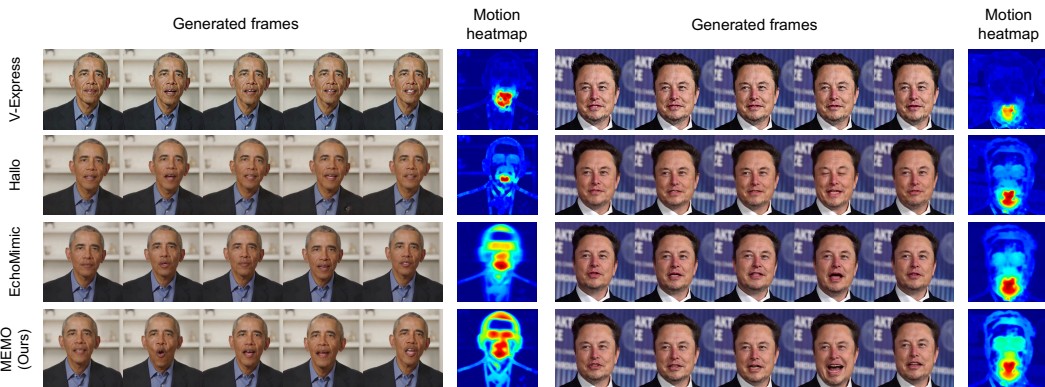

Figure 7: MEMO can generate talking videos featuring a wider range of smooth head movements and more emotional facial expressions, illustrated in both visualization and heatmaps.

## 5.3 QUALITATIVE RESULTS

**Comparisons with baselines.** Figure 5 presents several comparisons of talking videos generated by MEMO and the baselines on the two datasets we sampled. For the VoxCeleb test set, although existing methods can generate relatively realistic talking videos, their motion smoothness and expression-audio alignment are not satisfying compared to the ground truth videos. Compared to existing methods, our method can generate more natural facial expressions and head movements that are well-aligned with the audio inputs. In addition, the videos generated by MEMO have higher overall visual quality and better identity consistency. The advantages of MEMO are more significant in OOD datasets. Specifically, most existing models tend to generate artifacts and lose the original identity and details given reference images with pure background, as shown on the right of Figure 5. In contrast, MEMO can generate videos with similar quality compared to the ground truth.

**Diverse expression and head motion.** Figure 7 showcases the diversity in head motion and facial expressions generated by MEMO. This diversity enhances the naturalness and expressiveness of the talking videos. In addition to improvements in expressions and motions, our method also achieves better audio-expression alignment and audio-lip synchronization, as further shown by the human studies in Section 5.2. Video demonstrations can be found in the supplementary materials.

**Generalization to different scenarios.** To demonstrate the generalization capabilities of our method, we evaluate it under various challenging scenarios, *e.g.*, audios for singing and multiple languages, and reference images of virtual avatars. As shown in Figure 8, our method effectively generates lip movements synchronized with given singing voices. Furthermore, the model generalizes across different languages, producing accurate lip movements irrespective of the linguistic content. Additionally, we evaluate performance on images with diverse artistic styles, and our method maintains consistent quality across these variations. Video demos can be found in the supplementary materials.

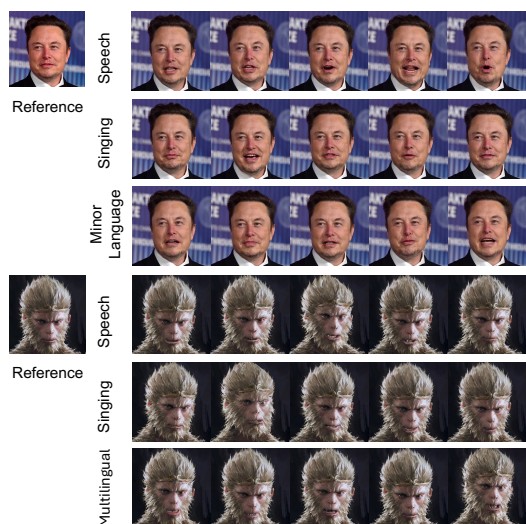

Figure 8: Visualization of generated videos on the Vox-Celeb2 test set and the OOD data reference images and audios. MEMO can generate talking videos with

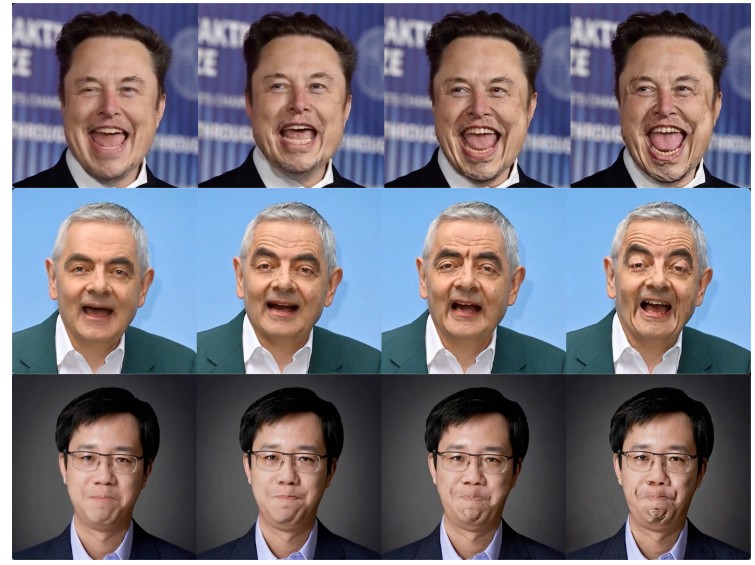

scale 2      scale 3      scale 5      scale 7

Figure 9: Ablation of the classifier-free guidance scale.

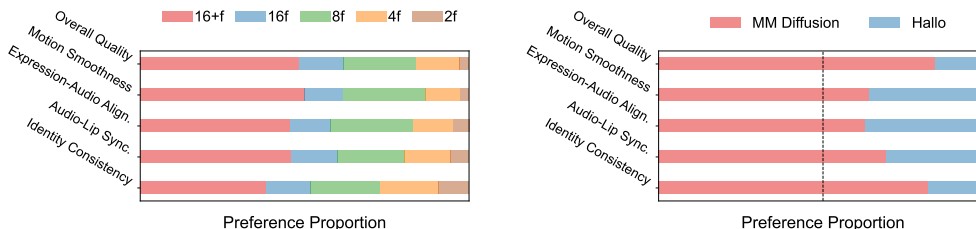

Figure 10: Human ablations on the number of past frames (*f*).

Figure 11: Human preferences between multi-modal attention and cross attention.

### 5.4 ABLATION STUDIES

**Effects of memory module.** We evaluate the effect of memory and the role of different past frame lengths as temporal guidance on video quality through human evaluations. As shown in Figure 10, longer memory significantly improves temporal coherence, overall quality, motion smoothness, identity consistency, and audio-lip alignment, while short motion frames leads to the worse performance. This result further demonstrate the effectiveness of our memory-guided temporal module.

**Effects of emotion guidance.** By adjusting the classifier-free guidance scale, we observe variations in the expressiveness of the generated faces. As shown in Figure 9, higher guidance scales lead to more pronounced emotional expressions. These visualization further verifies the effectiveness of our emotion-aware audio module.

**Effects of multi-modal attention.** We further investigate the impact of the multi-modal attention through human evaluations. Results in Figure 11 underscore the effectiveness of multi-modal attention over cross attention in terms of the overall video quality and lip-audio alignments.

## 6 CONCLUSION

In this work, we present MEMO, a state-of-the-art talking video generation model. MEMO reduces artifacts and error accumulation in long-term video generation by introducing the memory-guided temporal module. It can generate videos with high audio-lip synchronization and natural head movements with our emotion-conditioned audio module. In particular, it does not need face-related inductive biases in the model architecture, allowing it to be extended to broader applications, such as talking body generation tasks. In future work, it would be interesting to explore the effectiveness of Diffusion Transformer (Peebles & Xie, 2023) in talking video generation.

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

# APPENDIX

## A AUDIO EMOTION DETECTION

To facilitate emotion guidance in talking videos, it is crucial to develop an accurate and robust emotion detection module that can extract emotion labels from audio. Emotion recognition from speech and music has been extensively researched. Drawing on the well-established foundations of Speech Emotion Recognition (SER) and Music Emotion Recognition (MER), we aim to integrate these insights into a unified module.

### A.1 DATA

**Dataset collection**. To achieve robust emotion detection across both speech and music audio sources, we collected a large-scale dataset encompassing both speech and music segments, each annotated with emotion labels. A detailed overview of the datasets used in our training process is provided in Table 2. For the speech component, we sourced data from a recent Speech Emotion Recognition benchmark, EmoBox (Ma et al., 2024a), which incorporates 23 datasets from various origins, covering 12 distinct languages. Regarding the music component, we gathered data from the RAVDESS-song (Livingstone & Russo, 2018) and MTG-Jamendo (Bogdanov et al., 2019) datasets, including songs with and without background music.

| Speech Emotion Recognition datasets | | | | | |
|---|---|---|---|---|---|
| **Dataset** | **Source** | **Lang** | **#Emo** | **#Utts** | **#Hrs** |
| AESDD (Vryzas et al., 2018) | Act | Greek | 5 | 604 | 0.7 |
| ASED (Retta et al., 2023) | Act | Amharic | 5 | 2474 | 2.1 |
| ASVP-ESD(Landry et al., 2020) | Media | Mix | 12 | 13964 | 18.0 |
| CaFE (Gournay et al., 2018) | Act | French | 7 | 936 | 1.2 |
| EMNS (Noriy et al., 2023) | Act | English | 8 | 1181 | 1.9 |
| EmoDB (Burkhardt et al., 2005) | Act | German | 7 | 535 | 0.4 |
| EmoV-DB (Adigwe et al., 2018) | Act | English | 5 | 6887 | 9.5 |
| Emozionalmente (Catania, 2023) | Act | Italian | 7 | 6902 | 6.3 |
| eNTERFACE (Martin et al., 2006) | Act | English | 6 | 1263 | 1.1 |
| ESD (Zhou et al., 2021) | Act | Mix | 5 | 35000 | 29.1 |
| JL-Corpus (James et al., 2018) | Act | English | 5 | 2400 | 1.4 |
| M3ED (Zhao et al., 2022) | TV | Mandarin | 7 | 24437 | 9.8 |
| MEAD (Wang et al., 2020a) | Act | English | 8 | 31729 | 37.3 |
| MESD (Duville et al., 2021) | Act | Spanish | 6 | 862 | 0.2 |
| Oreau (KERKENI et al., 2020) | Act | French | 7 | 434 | 0.3 |
| PAVOQUE (Steiner et al., 2013) | Act | German | 5 | 7334 | 12.2 |
| Polish (Kaminska et al., 2015) | Act | Polish | 3 | 450 | 0.1 |
| RAVDESS (Livingstone & Russo, 2018) | Act | English | 8 | 1440 | 1.5 |
| SAVEE (Jackson & Haq, 2014) | Act | English | 7 | 480 | 0.5 |
| SUBESCO (Sultana et al., 2021) | Act | Bangla | 7 | 7000 | 7.8 |
| TESS (Dupuis & Pichora-Fuller, 2010) | Act | English | 7 | 2800 | 1.6 |
| TurEV-DB (Canpolat et al., 2020) | Act | Turkish | 4 | 1735 | 0.5 |
| URDU (Latif et al., 2018) | Talk show | Urdu | 4 | 400 | 0.3 |
| Music Emotion Recognition datasets | | | | | |
| **Dataset** | **Source** | **Lang** | **Emo** | **#Utts** | **#Hrs** |
| RAVDESS-Song (Livingstone & Russo, 2018) | Act | English | 6 | 1012 | 1.31 |
| MTG-Jamendo (Bogdanov et al., 2019) | Media | Mix | 56 | 5022 | 299.47 |

Table 2: Emotion Detection Dataset Information Table

We provide detailed information about each dataset in several aspects in Table 2: **Source** represents the origin of the samples, **Lang** specifies the dataset's language, **Emo** indicates the number of emotion categories, **Utts** shows the total number of utterances, and **Hrs** represents the total hours of training data. All data underwent a standardized processing protocol, being converted to a monophonic format with a sampling rate of $16,000$ Hz. Each utterance is uniquely annotated with an emotion label. For datasets containing lengthy samples, such as MTG-Jamendo, we divided them into shorter segments of 30 seconds to align with the typically shorter length of other datasets, as-

signing the same label to all segments. Each dataset was then split into training and testing sets with a ratio of $3 : 1$.

**Label merging**. A major challenge in integrating different datasets is aligning their label spaces, as each dataset often features distinct emotion categories. For instance, the URDU dataset (Latif et al., 2018) contains only four emotion labels: happy, sad, angry, and neutral. In contrast, ASVP-ESD (Landry et al., 2020) includes 12 emotion labels, covering less common emotions such as boredom and pain. For music emotion recognition datasets like MTG-Jamendo (Bogdanov et al., 2019), there are 56 mood/theme tags, not all of which correspond to emotional labels, and each sample can be assigned multiple tags. These discrepancies and overlaps in category spaces across different datasets present significant challenges for emotion detection.

To establish a generalized and streamlined label space, we designed our module to perform an 8-class classification task, selecting labels that are both commonly recognized and easily distinguishable: `angry`, `disgusted`, `fearful`, `happy`, `neutral`, `sad`, `surprised`, and `others`. We meticulously reviewed and mapped the original labels from each dataset to fit within this new label space. For instance, samples labeled as `pleasure` in the ASVPESD dataset were mapped to the `happy` category due to their semantic similarity. Labels that did not clearly correspond to a specific emotion were categorized under the `others` label.

## A.2 AUDIO EMOTION DETECTOR

We implemented an 8-way classifier for our task, drawing inspiration from state-of-the-art methods in speech and music emotion detection. Our solution builds upon Emotion2vec (Ma et al., 2024b), a robust universal speech emotion representation model. The feature extractor employs multiple convolutional layers and Transformer blocks and is trained using a teacher-student online distillation self-supervised learning approach. The feature extractor backbone of Emotion2vec is pre-trained on a large-scale multilingual speech corpus. For our classification task, we utilized the fixed Emotion2vec backbone as the feature extractor and trained a 5-layer MLP as the classification head.

To stabilize the training process, we applied gradient clipping, constraining the gradient updates within an $l_2$ norm of 1.0. To enhance the model's generalization ability, we incorporated a contrastive learning technique (Zhang et al., 2021a). The test accuracy for each dataset, as well as the overall accuracy, is reported in the table below. We compare with the original solution of Ma et al. (2024b) as the baseline, where they adopted a single linear layer after the feature extraction backbone for the downstream emotion detection task.

Figure 12: Accuracy comparison of audio emotion detection between Emotion2vec (Ma et al., 2024b) and our learned emotion detector.

| Dataset | Emotion2vec | Ours |
|---|---|---|
| AESDD | 75.84 | 78.52 |
| ASED | 86.20 | 85.23 |
| ASVP-ESD | 52.55 | 55.99 |
| CaFE | 73.30 | 100.00 |
| EMNS | 57.98 | 61.87 |
| EmoDB | 88.41 | 100.0 |
| EmoV-DB | 77.84 | 91.22 |
| Emozionalmente | 66.61 | 71.02 |
| eNTERFACE | 28.21 | 32.05 |
| ESD | 94.83 | 99.94 |
| JL-Corpus | 71.92 | 100.00 |
| M3ED | 42.59 | 41.52 |
| MEAD | 61.74 | 71.45 |
| MESD | 40.65 | 41.12 |
| Oreau | 50.96 | 42.31 |
| PAVOQUE | 85.15 | 92.74 |
| Polish | 44.89 | 100.00 |
| RAVDESS | 82.36 | 100.00 |
| SAVEE | 83.33 | 100.00 |
| SUBESCO | 78.43 | 100.00 |
| TESS | 76.29 | 95.14 |
| TurEV-DB | 47.45 | 53.47 |
| URDU | 54.00 | 56.00 |
| RAVDESS-Song | 43.58 | 100.00 |
| MTG-Jamendo | 65.30 | 74.50 |
| **Total** | **68.78** | **78.26** |

## B  DATA PROCESSING PIPELINE

We collect a comprehensive set of open-source datasets, such as HDTF (Zhang et al., 2021b), VFHQ (Xie et al., 2022), CelebV-HQ (Zhu et al., 2022), MultiTalk (Sung-Bin et al., 2024), and MEAD (Wang et al., 2020b), along with additional data we collected ourselves. The total duration of these raw videos exceeds 2,200 hours. However, as illustrated in Figure 13, we find that the overall quality of the data is poor, with numerous issues such as audio-lip misalignment, missing heads, multiple heads, occluded faces by subtitles, extremely small face regions, and low resolution. Directly using these data for model training results in unstable training, poor convergence, and terrible generation quality.

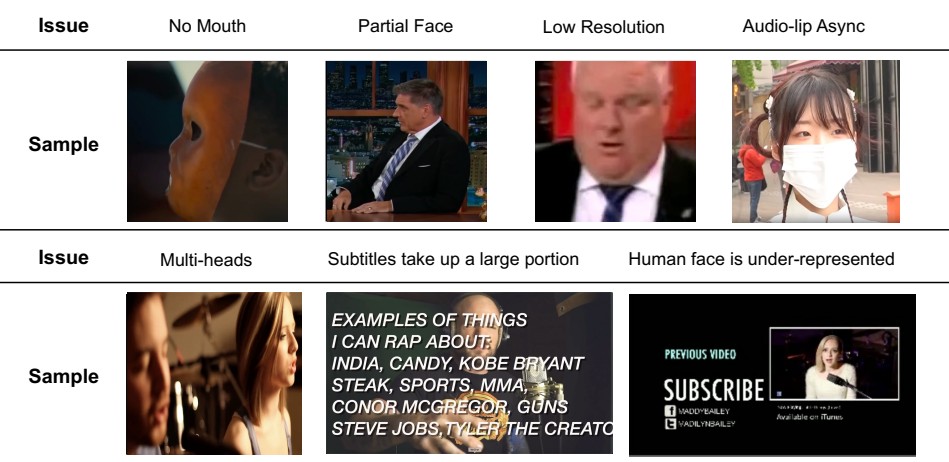

Figure 13: There are some issues making the dataset can't use in training since the training data needs to appear mouthy and the lips of the sound are consistent.

To further obtain high-quality talking head data, we developed a dedicated data processing pipeline for talking head generation. The pipeline consists of five steps: First, we perform scene transition detection and trim video clips to a length of less than 30 seconds. Second, we apply face detection, filtering out videos with no faces, partial faces, or multiple heads, and use the resulting bounding boxes to extract talking heads. To ensure that the cropped areas encompass more than just the human faces, we apply a scaling factor of 1.1 to the bounding box regions. Third, we use an Image Quality Assessment model (Su et al., 2020) to filter out low-quality and low-resolution videos. We apply an Image Quality Assessment (IQA) model to the first frame of the videos and find that when the IQA score exceeds 40, there is a noticeable improvement in video quality. Therefore, we use an IQA score of 40 as a selection criterion, but this threshold will be dynamically adjusted based on the volume and quality of the data. Fourth, we apply SyncNet (Prajwal et al., 2020) to remove videos with audio-lip synchronization issues. We use Sync-Confidence (Sync-C) to filter the data and find that it exhibits better diversity compared to Sync-Distance (Sync-D). Specifically, for a given dataset, not all data points tend to fall within the same scoring range, as shown in Figure 14. Additionally, for Sync-C, we can identify a suitable threshold for filtering, which is set at a score of 5 or higher. Lastly, for partial data, we manually assess the audio-lip synchronization and overall video quality for more accurate filtering. After completing the entire pipeline, the total duration of the processed high-quality videos is approximately 660 hours.

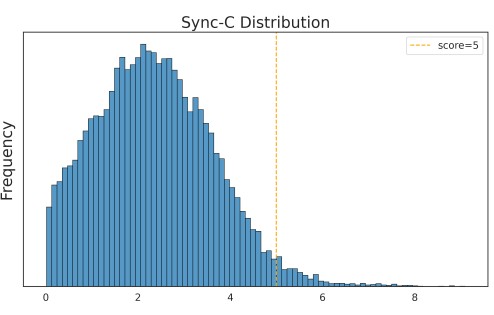 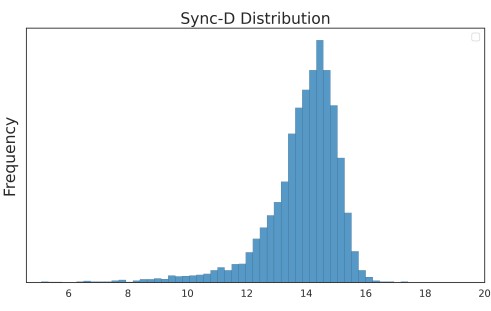

(a) Distribution of the Sync-C in the CelebV-HQ.  (b) Distribution of the Sync-D in the CelebV-HQ.

Figure 14: From the distributions of Sync-C and Sync-D, we observe that for the same dataset, the distribution of Sync-C is more dispersed, which facilitates the selection of an appropriate filtering threshold.

## C  MORE RELATED STUDIES OF DIFFUSION MODELS

Diffusion models (Sohl-Dickstein et al., 2015; Ho et al., 2020) are highly expressive generative models, demonstrating remarkable capabilities in image synthesis (Rombach et al., 2022; Podell et al., 2023) and video generation (Guo et al., 2023; Xing et al., 2023). Rombach et al. (2022) employ a UNet architecture and generate high-resolution images in the latent space, which is extended to video domains by AnimateDiff (Guo et al., 2023) via adding temporal attention layers. These models generate images or videos based on text prompts, where the text guidance from the pre-trained text encoder is introduced through cross-attention modules. In the domain of talking head, diffusion models also show promising results in generation quality (He et al., 2023; Tian et al., 2024; Wei et al., 2024; Xu et al., 2024a; Stypułkowski et al., 2024; Xu et al., 2024b), outperforming previous GAN-based methods (Prajwal et al., 2020; Zhou et al., 2020). Instead of using text prompts, most of these diffusion-based methods condition diffusion models on image and audio embeddings extracted from a pre-trained image encoder and audio encoder, respectively.

## D  MORE IMPLEMENTATION DETAILS

Both the Reference Net and the spatial module of the Diffusion Net are initialized with the weights of SD 1.5 (Rombach et al., 2022). The temporal module is initialized with the motion module from AnimateDiff (Guo et al., 2023). We add two projection modules to convert the audio embedding and image embedding into the dimensions required by our attention module. The audio embedding consists of all the hidden states from the Wav2Vec 2.0 model (Baevski et al., 2020). For both the Reference Net and the Diffusion Net, we replace the text cross-attention with image cross-attention. We use the normalized hidden states from the Reference Net before the self-attention layers for reference attention with the hidden states in the Diffusion Net. The training videos are center-cropped and resized to a resolution of $512 \times 512$ pixels. Across all training stages, we maintain a fixed learning rate of 1e-5. We train MEMO for 15k, 500k, and 100k steps at training stage 1, 2, and 3, respectively. During training, emotion embeddings are randomly dropped with a dropout probability of 30%, while all other conditions, including reference images, audio embeddings, past frames, are dropped with a probability of 5%. At inference, we set the frame rate to 30 FPS and generate 16 frames per iteration. The scale of classifier-free guidance is default to 3.5.