# OpenReview forum: "MEMO: Memory-Guided and Emotion-Aware Talking Video Generation"
_ICLR.cc/2025/Conference — ICLR 2025 Conference Withdrawn Submission_

### Official Review · Reviewer_G7Ty · 2024-10-18

**Soundness:** 3
**Presentation:** 3
**Contribution:** 2
**Rating:** 5
**Confidence:** 5

**Summary:**

The proposed solution MEMO is an end-to-end system designed for creating identity-consistent and expressive talking videos using audio inputs. MEMO comprises two main components: (1) A memory-guided temporal module that maintains long-term identity consistency and ensures smooth motion. This is achieved by storing information from previously generated frames in memory states, which then guide the temporal modeling process using linear attention. (2) An emotion-aware audio module that improves the interaction between audio and video by employing multi-modal attention instead of traditional cross-attention methods. This module also detects emotions from the audio feed and uses them to adjust facial expressions via emotion adaptive layer normalization.

**Strengths:**

1. The performance of MEMO is decent.
2. The workload of this paper is considerable, ranging from data collection and processing to training.

**Weaknesses:**

1. Incremental and limited novelty. Overall, the method in this paper adds incremental and limited new modules to the architecture of the EMO[1] approach. This paper argues that the previous approach using emotion labels is insufficient, but the emotion module used in MEMO essentially still labels each data segment with emotions through audio. Moreover, predicting emotions from audio may not be as accurate as using video, especially on the MEAD dataset. Therefore, the emotion module feels somewhat meaningless. Although the memory module does indeed improve inter-frame continuity as stated in the paper, it feels somewhat incremental.
2. Insufficient evaluation for emotion module. The paper does not effectively evaluate whether the emotion module can generate facial expressions consistent with the audio emotion. Fig. 9 claims that the emotion module can generate such expressions, but in the example of "musk," the audio sounds angry, while the video does not seem anger. Additionally, since this paper involves emotions, it should discuss and compare emotional methods; however, the baselines compared in this paper are mostly under neutral emotion settings.
3. Data processing should not be considered a contribution. The data processing in this paper is very similar to GAIA[2] and is a basic procedure.

[1] Tian, Linrui, et al. "Emo: Emote portrait alive-generating expressive portrait videos with audio2video diffusion model under weak conditions." arXiv preprint arXiv:2402.17485 (2024).

[2] He, Tianyu, et al. "Gaia: Zero-shot talking avatar generation." arXiv preprint arXiv:2311.15230 (2023).

**Questions:**

Could you please further evaluate the effectiveness of the emotion module.

---

### Official Review · Reviewer_Motj · 2024-10-31

**Soundness:** 2
**Presentation:** 3
**Contribution:** 3
**Rating:** 6
**Confidence:** 3

**Summary:**

This paper proposes MEMO, an improved diffusion-based method for emotion-aware talking face generation. Authors introduce the memory guided temporal module that allows long term identity consistency efficiently using linear attention. They replace the cross-attention module with a dynamic multi-modal attention module. They incorporate the dynamically detected emotion (from audio) using emotion-adaptive LayerNorm. Further, a data processing pipeline is introduced to filter high quality training data from multiple datasets (HDTF, VFHQ, CelebV-HQ, MultiTalk, MEAD and some collected videos). Along with VoxCeleb2, a nice evaluation is performed on out-of-dist data. Human evaluation is also included along with some good ablations.

**Strengths:**

1) An improvement over current diffusion-based talking face generation methods. Memory-guided temporal module is the main novelty. The multimodal attention module (combining the representations of audio and video), and the emotion-adaptive layer norm further add to the novelty. All along authors use already existing successful methods like Linear Attention, Adaptive Normalization, classifier-free guidance, rectified flow loss etc to make their work efficient and better.
2) Dynamic past frame training is an intelligent approach to adapt to longer past frames during inference.
3) Effort to filter the datasets using a new data processing pipeline and merging of emotion labels is appreciated. Talking face datasets with emotion labels are limited, and every dataset has some biases. So combining these datasets and filtering them can be really helpful in the training process.
4) Paper has a strong evaluation on Out-of-distribution dataset against relevant methods.
5) Ablation of classifier-free guidance for amount of emotion incorporation is interesting.
6) Supplementary material contains impressive results, especially on singing.

**Weaknesses:**

1. Paper claims that emotion "dynamically detected emotion from audio" is always better than "static emotion labels". Accuracy of emotion-classifier used will implicitly determine the quality of talking face generation, no such problem exist with using emotion labels. Moreover If you have want a talking video with emotion different from the audio available, then static emotion label can be a better choice.
2.  Evaluation metric to analyze the emotion incorporation is missing (something like matching emotions detected from audio input and video generated).
3. Results included in supp. material are impressive in singing and english-audio, but when the audio-language is not english, methods like EchoMimic seem to generate better video.
4. Computational analysis can be included for inference speed and memory. It can be really useful for method's applications.

**Questions:**

1. Will you make the dataset public? That will be helpful for community.
2. Please compare with more emotion-ware methods in the evaluation to better judge the emotion incorporation.
3. Its guessable from the paper that "how emotion is included in adaptive layer norm", Still including the exact method/equation will be helpful.

---

### Official Review · Reviewer_9NEa · 2024-11-02

**Soundness:** 2
**Presentation:** 3
**Contribution:** 2
**Rating:** 3
**Confidence:** 5

**Summary:**

This paper introduced memory-guided and emotion-aware talking face generation based on diffusion model. It proposed 2 module, a memory-guided temporal module and an emotion-aware multi-modal attention module, to solve seamless audio-lip synchronization, long-term identity consistency and natural expression problems. Experiments show better results compared to existing methods.

**Strengths:**

1) Novel approach to solve problems: Proposed modules, a memory-guided temporal module and an emotion-aware attention module, solve the problem properly.
2) Clear presentation: Well-written and presents findings in a clear and concise manner.
3) The way of preprocessing the dataset is meticulous.

**Weaknesses:**

1) There's no overall loss function. Besides equation 1, 5 and 6, are there any other loss functions such as image reconstruction loss (L1 loss, LPIPS)? The author should add an equation or detailed description of overall loss function to explain how it works.
2) Quantitative ablation study on memory module, emotion guidance, multi-modal attention is missed. Please add ablation results on these 3 modules using FVD, SSIM, Sync-C.
3) When comparing with Hallo, EchoMimic and other methods,  is their models re-trained with the same cleaned dataset as MEMO? If not, the improvement in image quality is likely due to the higher quality of the training data used. Taking No.2 into account, the proof of effectiveness of memory module, emotion guidance, multi-modal attention is rather weak. The author should explicitly state the data used for training each model and add the comparative experiments results using the same cleaned dataset for training.
4) No further study on error accumulation which is mentioned at the very start, such as how the memory module works to avoid error accumulation theoretically. And why the error accumulation would affect identity consistency with the Reference Net providing information of reference image. The author should provide visualization analysis between MEMO and other methods, or supplement quantitative measures of error accumulation over time with or without the memory module.
5) As we known, rectified flow presented in SD3 paper was solid, but it is suggested that the original paper of rectified flow should be updated as references. ('Flow Straight and Fast: Learning to Generate and Transfer Data with Rectified Flow, Xingchao Liu, Chengyue Gong, Qiang Liu'.)

**Questions:**

1) There is a strange phenomenon in Table 1. All the methods show better FVD and FID on OOD dataset over voxceleb2, except MEMO's FID. According to your manuscript, OOD dataset should be more difficult to handle. Is there any further explanation?
2) What's the meaning of MM diffusion in Figure 11? The ablation study on effects of multi-modal attention should be done between MEMO with MM-diffusion and MEMO without MM-diffusion.

---

### Note · Authors · 2024-11-13

I have read and agree with the venue's withdrawal policy on behalf of myself and my co-authors.